# Biological Graft as an Innovative Biomaterial for Complex Skin Wound Treatment in Dogs: A Preliminary Report

**DOI:** 10.3390/ma15176027

**Published:** 2022-09-01

**Authors:** Adriano Jaskonis Dall’Olio, Gustavo de Sá Schiavo Matias, Ana Claudia Oliveira Carreira, Hianka Jasmyne Costa de Carvalho, Thais van den Broek Campanelli, Thamires Santos da Silva, Mônica Duarte da Silva, Ana Lúcia Abreu-Silva, Maria Angélica Miglino

**Affiliations:** 1Surgery Department, School of Veterinary Medicine and Animal Science, University of São Paulo, São Paulo 05508-270, Brazil; 2Department of Veterinary Pathology, State University of Maranhão, Maranhão 65055-150, Brazil

**Keywords:** innovative wound treatment, canine skin, decellularized and recellularized scaffolds, regenerative medicine, tissue engineering

## Abstract

Complex wounds in dogs are a recurrent problem in veterinary clinical application and can compromise skin healing; in this sense, tissue bioengineering focused on regenerative medicine can be a great ally. Decellularized and recellularized skin scaffolds are produced to be applied in different and complex canine dermal wounds in the present investigation. Dog skin fragments are immersed in a 0.5% sodium dodecyl sulfate (SDS) solution at room temperature and overnight at 4 °C for 12 days. Decellularized samples are evaluated by histological analysis, scanning electron microscopy (SEM) and gDNA quantification. Some fragments are also recellularized using mesenchymal stem cells (MSCs). Eight adult dogs are divided into three groups for the application of the decellularized (Group I, *n* = 3) and recellularized scaffolds (Group II, *n* = 3) on injured areas, and a control group (Group III, *n* = 2). Wounds are evaluated and measured during healing, and comparisons among the three groups are described. In 30- and 60-day post-grafting, the histopathological analysis of patients from Groups I and II shows similar patterns, tissue architecture preservation, epithelial hyperplasia, hyperkeratosis, edema, and mononuclear inflammatory infiltrate. Perfect integration between scaffolds and wounds, without rejection or contamination, are observed in both treated groups. According to these results, decellularized skin grafts may constitute a potential innovative and functional tool to be adopted as a promising dog cutaneous wound treatment. This is the first study that applies decellularized and recellularized biological skin grafts to improve the healing process in several complex wounds in dogs, demonstrating great potential for regenerative veterinary medicine progress.

## 1. Introduction

Skin wounds in dogs present a high incidence in veterinary clinical–surgical routines [1], and can be caused by extrinsic factors, such as surgical incisions or trauma, and intrinsic factors, followed by infections, chronic ulcers, neoplasms, vascular and metabolic alterations [2,3,4]. Several events may influence the healing process, such as those related to the animal or associated with the wound characteristics, the chronicity, and the healing capacity of the injured tissue. Additionally, external factors can also lead to complications during the healing process, prolonging it [5,6,7].

Extensive skin defects require complex management, including necrotic tissue debridement, local and/or systemic infection control, and tissue regeneration induction [8]. Multiple therapeutic resources have been studied, aiming to advance and accelerate the skin healing process [9,10,11]. The great challenge for cutaneous repair is the lack of an adjacent tissue for the closure and healing of large extension wounds [12]. These events limit conventional therapeutic approaches and, in diverse situations, make them ineffective [13].

Skin grafts are largely applied for tissue repair and are classified as: autogenic (which are taken from the individual’s own body, with no chance of immune rejection) [14,15], xenogeneic (taken from a different species, but commonly leading to immune rejection and disease transmission) [16,17], and allogeneic (taken from different individuals of the same species; this is often required in large wound cases, but can also lead to immune rejection) [18].

Allotransplantation is a clinical reality in plastic and reconstructive surgery [19]; unfortunately, the survival of allogeneic composite tissue depends on the use of chronic nonspecific and novel specific immunosuppressive treatment, although many of them carry a risk for neoplasms, opportunistic infections, and/or end-organ toxicity [20]. A decellularized composite matrix created by tissue engineering has progressed in recent years. With medical bioengineering, it is possible to appoint the decellularization process as a promising technique promoting cell removal, while preserving the native extracellular matrix (ECM) [21,22,23]. Bioscaffold ECM preservation can act as an ideal environment for sustenance, proliferation, and cell distribution, serving as a mechanical support, increasing the bioavailability of tissues and organs, and reducing rejection after transplants [24].

ECM contains growth factors and collagen fibers required for ECM protein deposition, angiogenesis, and epithelialization [25,26]. It also confers elasticity and resistance to the skin tissue, as it is mainly composed of collagen types (I and III), glycosaminoglycans (GAGs), proteoglycans, adhesive glycoproteins (fibronectin, laminin and vitronectin), and elastic fibers [27,28]. The preservation of these components after cell removal is essential to give the necessary support for cell proliferation, differentiation, and tissue restoration. Vascular structure preservation is also possible with the ECM maintenance, which can improve healing process after grafting or transplantation [29,30,31,32,33].

This investigation aims to evaluate the efficiency of two options of canine skin scaffolds: decellularized and recellularized scaffolds with adipose tissue-derived mesenchymal stem cells (ADMSCs) grafted to treat complex skin wounds. Biomaterials are a sustainable alternative; they are low cost and offer an appropriate innovative therapeutic approach for reconstructive regenerative medicine. The relevance of this study is based on the application, for the first time, of this biological graft technology in dogs presenting lesions from different causes, mimicking common cases in veterinary clinical practice. Since, for ethical reasons, these lesions cannot be intentionally induced in dogs for a standardized study, it was decided to use clinical cases.

New options are needed for extensive wound treatments in canines since the current methods are expensive and require a high demand for specialized personnel to establish protocols and perform dressings. In this way, this proposed treatment requires easily obtained material, covering the affected area, and showing greater skin regularity and restored tissue.

## 2. Materials and Methods

### 2.1. Ethics Approval

This experiment was approved by the Ethics Committee on Animal Use of the School of Veterinary Medicine and Animal Science of the University of São Paulo (Protocol Number 1733150419) and the Veterinary School Hospital of Jaguariúna University Center (Protocol Number 019/2019).

### 2.2. Skin Fragment Collection

Abdominal skin fragments (ventral region) with an area of 10 cm^2^ and a thickness of 0.5 cm from six fresh dog cadavers of both genders, ranging from 2 to 7 years, were used. Fragments were dissected from each animal and kept frozen at −80 °C until the decellularization process.

### 2.3. Skin Decellularization

The decellularization process was performed at the Veterinary Regenerative Medicine Laboratory, School of Veterinary Medicine and Animal Science, University of Sao Paulo, Brazil. Skin fragments were washed in distilled water, followed by 1% tetrasodium ethylenediaminetetraacetic acid solution (EDTA, #E2005.01.AG Synth) and 0.5% antibiotics (penicillin/streptomycin, LGCBio, Cotia, Brazil), for 5 min under immersion and orbital agitation at 100 rpm. The fragments were also placed in 0.5% sodium dodecyl sulfate (SDS, #13-1313-01, LGCBio) at room temperature during the day and overnight at 4 °C for 12 days. Finally, the fragments were washed in 1% Triton X-100 (#13-1315-05, LGCBio) 3 times for 5 min.

### 2.4. Scaffolds’ Histological Analysis

Small fragments (0.5 cm^2^) of native and decellularized samples were used to perform the ECM histological analysis. Each fragment was fixed in 4% paraformaldehyde (PFA) for 48 h, dehydrated with increasing concentrations of alcohol (70, 80, 90 and 100%) for 20 min each, diaphanized in xylene, and included in paraffin. Then, 5 μm microsections (#RM2265, Leica, Wetzlar, Germany) were transferred to slides and stained by hematoxylin–eosin and Masson’s Trichrome techniques. Finally, slides were analyzed and photographed under a light optical microscope (FV1000 Olympus IX91, Tokyo, Japan) at the Advanced Diagnostic Imaging Center (CADI-FMVZ/USP).

### 2.5. DAPI Fluorescence

Decellularized samples were frozen in a −150 °C freezer, embedded in an optimal cutting temperature solution (O.C.T. Sakura Fineetek 4583, Torrance, CA, USA) and sectioned in a cryostat (LEICA, CM1860) at −30 °C. Microsections of 10 μm were transferred to frosted-edged glass slides, thawed at room temperature for 15 min, and stained with 4′,6′-diamino-2-phenyl-indole (DAPI) solution [1:10,000] in 1× phosphate-buffered saline (PBS) for 10 min in the dark. Then, the slides were washed with distilled water, observed, and photographed under a fluorescence microscope (NIKON Eclipse 80i) to verify the presence or absence of cellular nuclei.

### 2.6. Scaffolds’ Ultrastructural Analysis

In order to analyze the extracellular matrix structure, native and decellularized fragments (0.5 cm^2^) were processed with scanning electron microscopy (SEM). The samples were fixed in 4% PFA for 72 h and washed 6 times for 5 min each in an ultrasonic bath (USC1450-UNIQUE) with distilled water. After that, the fragments were dehydrated with increasing alcohol concentrations (70, 80, 90, and 100%) for 10 min each, and transferred to the critical point camera (LEICA EM CPD 300*). After 40 min, the fragments were adhered to stubs with double-sided carbon tape, (#K550-Emitech, Ashford, UK) for gold metallization and analysis by SEM (LEO 435 VP^®^).

### 2.7. gDNA Quantification

Native and decellularized samples (25 mg) were used to quantify the remaining genomic DNA with the DNA Mini Kit QIAamp^®^ (Qiagen, Hilden, Germany), according to the manufacturer’s specifications. Samples were digested overnight at 56 °C with proteinase K and a lysis buffer kit, purified, and analyzed by spectrophotometry at 260 nm (Nanodrop, Thermo Scientific, Waltham, MA, USA).

### 2.8. Biological Scaffold Recellularization

Canine ADMSCs, which were previously characterized and donated by Carreira’s group [34], were used for scaffold recellularization. The decellularized skin fragments were washed in 1× PBS solution with 0.5% antibiotic (penicillin–streptomycin, LGC Bio, Hoddesdon, UK), and kept under ultraviolet light (UV) for 10 min for sterilization before recellularization. The sterility was tested in Alpha-MEM culture medium and incubated at 37 °C with 5% CO_2_ for 24 h. Then, 5 × 10^4^ canine AD-MSCs were placed into sterilized decellularized scaffolds in untreated plates (Sarstedt, Newton, NC, USA) for 7 days of culture [32]. Analysis of the recellularized scaffold was performed by SEM, as described above.

### 2.9. Untreated and Treated Animal Groups

Eight canine patients from the clinical routine of the Jaguariúna Veterinary School Hospital were selected for the present investigation. These dogs presented complex wounds resulting from different causes. They had no other clinical complications and chronic diseases besides the wounds and were divided into 3 groups. In Group I (*n* = 3), animals were treated by grafting the decellularized skin scaffolds on the injured areas. Group II (*n* = 3) animals were treated by grafting canine recellularized skin with AD-MSCs on the injured areas. Group III (*n* = 2) animals were used as a control and submitted for wound treatment by second intention (Table 1). Information of each case is described in Table 1.

### 2.10. Preoperative and Wound Debridement

All animals were submitted to physical examination, electrocardiogram, and hematologic and biochemical analysis (blood cell count, urea, alkaline phosphatase, alanine aminotransferase and blood glucose) in order to verify any cardiac, renal function or coagulation disorders ( Appendix A). In order to perform the wound debridement, animals from the treated groups (I and II) were submitted to pre-anesthesia using acepromazine (0.1 mg/kg, intramuscular) associated with Tramadol (2 mg/kg, intramuscular). After 15 min, the anesthesia was induced with propofol (5 mg/kg, intravenous), with subsequent endotracheal intubation and anesthetic maintenance with isoflurane. Wounds were debrided, including the removal of non-viable tissue, cellular debris, and all foreign debris to minimize wound infection and promote wound healing.

Appendix A shows the electrocardiogram, hematological and biochemical results of the dogs before scaffold grafting.

### 2.11. Scaffold Grafting Technique

Ten days after wounds debridement, the injured areas were cleaned with 0.5% chlorhexidine solution. Patients were anesthetized again using the same anesthetic protocol for the scaffold’s grafting previously mentioned. Staggered mechanical slits (5 to 15 mm in length and 2 to 6 mm in distance) were performed parallel to the skin tension lines for better exudate drainage, facilitating graft adherence to ensure complete wound coverage. The scaffolds were fixed in the target areas with a separate simple suture pattern, using a 3-0 nylon suture thread (SHALON^®^, Osaka, Japan). 3 to 4 mm was determined between the stitches, bringing the edges of the grafted scaffold closer to the wound edges, as suggested by Paim [35] and Radlinsky [36] (Appendix A). After grafting, non-adherent padded bandages were applied on the grafted areas with low compression to favor the integration between the scaffold and the wound, in addition to reducing hematoma and seroma occurrence, with the consequent loss of tissue viability [3,37]. Bandages were changed daily in an absorbent and non-adherent manner. The patients remained for 10 days in a restricted area, using an Elizabethan collar. Histopathological and statistical analyses were performed in order to evaluate the viability of the grafted scaffold to the healing process.

### 2.12. Statistical Analysis

For grafting healing evaluation, the following parameters were considered: the evaluation of the wound initial area (cm^2^) and the maximum time (days) for complete tissue repair by obtaining the diameters of the lesions. The results were expressed through regression analysis, Duncan’s test, and analysis of variance with values (*p* < 0.05) for those that showed statistical differences.

## 3. Results

### 3.1. Canine Skin Decellularization Analysis

Decellularized scaffolds showed a translucid aspect, as well as a discrete white color, evidencing cell removal and MEC preservation (Figure 1).

Histological slides, stained by hematoxylin–eosin (HE), Masson’s trichrome (MT) (Figure 2A,B), and DAPI fluorescence (Figure 2D) techniques, of the native tissue showed cell nuclei presence and collagen fibers arranged and stained in blue (Figure 2A,B), while the SEM analysis showed the ultrastructure organization of the native tissue (Figure 2C).

HE and DAPI staining showed the absence of cell nuclei in the decellularized scaffolds (Figure 2E,H), and preserved collagen fibers stained in blue by the MT technique (Figure 2F). Collagen fibers remained evident through SEM analysis, as did matrix organization (Figure 2G).

### 3.2. Genomic DNA Quantification

The genomic DNA concentration was significantly reduced from 62.9 ± 13.86 ng/mg in the native skin to 1.94 ± 1.3 ng/mg of tissue in the decellularized scaffold (Figure 3).

### 3.3. Analysis of Recellularized Scaffolds

Scaffolds recellularized with canine AD-MSCs were evaluated by SEM analysis. Through the obtained images, it was possible to observe the cell growth and its adherence to the scaffolds (Figure 4).

### 3.4. Pre- and Post-Grafting Macroscopic Analysis

The wounds of each dog from Group I were macroscopically described (Appendix A) before and after scaffold grafting. Figure 5A,G,M depict the wounds of each patient before scaffold grafting as soon as they were admitted to the Veterinary Hospital and 10 days after the cleaning and debridement process (Figure 5B,H,N), followed by the scaffold implantation (Figure 5C,I,O) and the follow-up after 48 h (Figure 5D,J,P), 10 days (Figure 5E,K,Q), and the day of complete wound healing after scaffold grafting (Figure 5F,L,R).

After the debridement process, a cleaner wound and healthy granulation tissue were observed immediately scaffold grafting. In dogs 1 and 3, the scaffold was completely integrated into the wound within 48 h (Figure 5D,P), while in dog 2 (Figure 5J) the start of integration was observed between the scaffold and the wound at 48 h. The scaffold remained in the wound bed for a longer period compared to dogs 1 and 3. On the 10th day, all dogs from this group showed a wound size reduction and edge retraction, in addition to the beginning of re-epithelialization. In this group, complete wound healing and fur growth took 31 to 45 days. 

Additionally, the wounds of each dog from Group II were macroscopically described (see Appendix A) before and after grafting the recellularized scaffold (Figure 6). It was observed that the grafted recellularized scaffold (Figure 6C,I,O) partially adhered to the wound within 48 h (Figure 6D,J,P). After 10 days, the wound showed a decrease in size, beginning with granulation tissue formation (Figure 6E,K,Q). The complete healing time of this group was 45 to 50 days (Figure 6F,L,R). The wounds of dogs 4 and 6 had the same healing time. The wounds of dogs 5 and 6 had the same healing time.

Finally, the healing wounds of Group III were completed by second intention and each wound was described (see Appendix A) and photographed before and after the treatment (Figure 7). Within 10 days of healing (Figure 7C,H), it was possible to observe granulation tissue formation with partial wound closure in wounds of dogs from this group. Dog 7 had the smallest wound size among all the animals included in this study and, therefore, complete healing was faster, lasting only 24 days (Figure 7E), while the wound of dog 8 took 35 days to reach total healing (Figure 7J).

### 3.5. Microscopic Analysis


*Histopathological descriptions of Group I (30 days post-grafting)*


Dog 1’s histopathological analysis showed discrete melanocytic cellular hyperplasia and hyperkeratosis in the epithelial region. The dermis showed discrete mast cell presence and moderate edema foci represented by the spacing of collagen fibers (Figure 8A^I^,A^II^). Unfortunately, an evaluation of dog 2 was not possible due to the COVID-19 pandemic (early 2020), which impaired the animal’s return to the veterinary hospital for clinical and biopsy procedures. Regarding dog 3, the epithelium was intact with discrete multifocal hyperplastic areas, disarranged collagen fibers, and discrete edema (Figure 8C^I^,C^II^).


*Histopathological descriptions of Group I (60 days post-grafting)*


Sixty days post-grafting, the dogs showed wound healing evolution. Dog 1 showed an intact epithelium with discrete hyperplasia and hyperkeratosis. The dermis presented reduced edema foci, discrete mast cells and capillary congestion presence (Figure 8A^III^,A^IV^). A hyperplastic epithelium was also observed in dog 2, in addition to a discrete fibrinoid material deposition, neoformed vessels, angiectasia, moderated hemorrhage foci, and discrete fibrosis, as well as pilous follicles, sweat and sebaceous glands (Figure 8B^I^–B^IV^). Similarly, dog 3 presented multifocal areas of irregular epithelial hyperplasia, discrete edema in the dermis, as well as discrete fibrosis and reactive fibroblast bundles (Figure 8C^III^,C^IV^).


*Histopathological descriptions of Group II (30 days post-grafting)*


Dog 4 presented moderate epithelial hyperkeratosis, discrete edema, neoformed vessels, angiectasia, discrete lymphoplasmacytic inflammatory infiltrate in perivascular areas, and discrete hemorrhage (Figure 8D^I^,D^II^). Dog 5’s epithelium showed discrete alterations, with edematous area, neoformed vessels, discrete hemorrhage and lymphoplasmacytic inflammatory infiltrate surrounding the glandular epithelial, which was evident in this animal (Figure 8E^I^,E^II^). Dog 6 showed hyperplastic cells followed by intradermic papillary projections, discrete edematous areas, discrete fibrosis mediated by fibroblastic bundles, discrete neoformed vessels and angioectasia (Figure 8F^I^,F^II^).


*Histopathological descriptions of Group II (60 days post-grafting)*


Dog 4 showed an intact epithelium, as well as discrete edematous areas, lymphoplasmacytic inflammatory infiltrate close to the basal membrane, and moderate angiectasia (Figure 8D^III^,D^IV^). Dog 5 also presented these discrete alterations, in addition to skin attachments such as hair follicles and glandular tissue (Figure 8^III^,E^IV^). Dog 6 showed discrete epithelial hyperplasia, neoformed vessels, discrete hemorrhage, and mononuclear inflammatory infiltrate (Figure 8F^III^,F^IV^).

### 3.6. Statistical Analysis—Complete Wound Healing

The mean area of each lesion was determined, and the measurements were correlated with the time required for the complete healing of the wound for each patient (Table 2).

It was observed that the wounds progressed continuously (regression analysis) when the wound areas and the time required for complete healing were analyzed. Through the Duncan’s test graphic, it was observed that Group II showed smaller wound sizes compared to Group I; however, a longer post-grafting healing period was also observed, which revealed a 0.19 R2 value (Figure 9). 

Through Duncan’s test, different results were obtained for the three applied treatments since the treatment and the size of the wound influenced the total healing time (Figure 10).

The treatment applied and the size of the wound significantly impacted healing time. The fitted line graph indicated a greater linear relationship between the treatments of Group I and the control group (Group III) with close scar aspects. The graph of the treatment line in Group II showed an individual scar aspect due to the distances of the results above the adjusted line, differing from the other groups (Figure 11).

## 4. Discussion

Complex skin damage may occur in many different clinical cases and the use of allogeneic decellularized skin scaffolds to reconstruct the damaged area has been successfully applied in plastic and reconstructive surgeries in human medicine [38]. The preserved ECM in decellularized scaffolds has structural properties that can provide a suitable environment for cell migration, growth, and replication to the injured area, improving the healing process, wound closure, and reducing scar tissue [39,40,41]. Large skin defects and complex wounds represent a serious problem for veterinary medicine; however, tissue banks and bioscaffold application in reconstructive surgeries for dogs are unusual [42,43]. This investigation proposes a new idea for skin wound treatment in dogs, addressing regenerative medicine in clinical practice. In this study, we describe the first preliminary report of decellularized and recellularized skin scaffold application in dog wounds attended in clinical routine as an innovative, simple, inexpensive, and efficient alternative reconstructive solution.

The skin is a tissue with high antigenicity, which can lead to rejection in both allograft and xenograft approaches [37,44,45]. Thus, tissue engineering technologies have been developed as biocompatible and biodegradable biomaterials to avoid rejections and promote effective healing by removing DNA components and pathogens [46]. The decellularization technique used in this work to obtain skin scaffolds preserved ECM collagen fibers, as observed through histological and SEM analysis. Moreover, this technique successfully promoted cell nuclei removal, as demonstrated by gDNA quantification.

Additionally, previous studies have shown the better reparative capacity of recellularized scaffolds, scaffolds, and stem cells (i.e., the adipose-derived stem cells (ASCs)) and isolated stem cells in wounds, stimulating cell proliferation and self-renovation of the injured tissue [47,48,49,50,51,52,53,54]. Previous studies in humans using ASCs in wounds have shown important antimicrobial activity without infection, crusts, or exudate production [55,56].

The healing process may differ according to the wound cause and characteristics, management, and comorbidities presented by the animal [57]. In our study, the dogs grafted with decellularized scaffold (Group I) presented full integration of the tissue within 48 h, except for dog 2. The wound of this dog was caused by a capybara bite, which was deeper and more contaminated; therefore, this may have delayed the healing process compared to dog 1 and impaired the full quick integration of the scaffold in the wound of this dog. Macroscopically, the wounds of Group I took less time for complete healing as well as having a better skin scar appearance. Additionally, Group I presented important histological alterations at 30- and 60- days post-grafting, including hyperkeratosis, discrete edema, and mast cells, but there was an absence of bacterial contamination or chronic suppurative inflammation, indicating the graft success even in dog 2. No histopathological characteristics of tissue malignancy were observed in dog 3, who presented a neoplastic formation.

On other hand, grafted recellularized scaffolds (Group II) presented a longer time integration, within 72 h, in comparison with Group I, as well as complete healing. This is evidenced by the statistical analysis in Figure 9, which presented a lower R2 value due to the patients of Group II. This may have happened due to the presence of cellular material in the Group II scaffold, which can promote a greater inflammation process and a longer healing time.

Interestingly, this group presented significant alterations analyzed by the histopathological method compared to animals from Group I. Although these lesions were determined by different causes, Group II descriptions were slightly homogeneous, with a predominance of 30- and 60-day post-grafting mild edema, hyperplastic keratinized stratified squamous epithelial tissue, neoformed vessels, and the mononuclear inflammatory process. Skin appendages, such as glandular tissue and hair follicles, were present, which was also clearly observed in the macroscopic analysis.

The immune response to antigenic components of xenogeneic or allogeneic tissues represents a critical barrier to the use of scaffolds in transplants [58], and even low levels of immune response to these products can compromise the function or cause the destruction of the transplanted tissue; thus, it is necessary to avoid residual antigenicity using cell removal [59,60]. In this sense, the macroscopic aspects and histopathological results observed in Group II, when compared to Group I, were probably related to the greater antigenic response of the animals to the cells in the recellularized scaffolds. This is possibly due the fact that, compared to a decellularized scaffold, absent of cells and DNA material, a recellularized scaffold has allogeneic cellular material that can lead to antigenic response and, consequently, influence inflammatory and hemorrhagic processes, even discrete ones. Furthermore, we demonstrated that both groups receiving decellularized and recellularized scaffolds, respectively, presented a good response compared to the control groups, and Group I presented better outcomes compared to Group II.

According to Broughton et al. [61], after a tissue injury, there is the release of vasoconstrictor substances by cell membranes and, in addition, the injured endothelium and platelets stimulate coagulation cascade. From this point, the inflammatory response begins with vasodilation and increased vascular permeability, promoting the neutrophils’ migration to the injured area [61]. Two to four days are necessary to observe macrophage migration to the lesion site, followed by their replication. This contributes to the initial debridement by neutrophils and contributes to angiogenesis, fibroplasia, and the extracellular matrix synthesis [61], which are fundamental elements for the transition to the proliferative phase. Due to the periods of histopathological examination (30- and 60- days post-grafting), it was impossible to clearly visualize these initial healing processes in the animals under study; the analyses were consequently limited to the macroscopic observations made during the healing process. 

In general, after injury, fibroblasts are activated and stimulated, producing type I collagen, and transforming it into myofibroblasts, promoting wound contraction [58]. From the 4th to the 14th days after the injury, the proliferative phase begins with the formation of granulation tissue, collagen deposition and angiogenesis; however, if the basement membrane is damaged, as in the case of the ulcerative areas, epithelial cells at the wound edges begin to proliferate, reestablishing the protective barrier and delaying the epithelialization process [62]. In this way, the rapid healing of the grafted patients in the present study can be associated with basement membrane continuity stimulation due to scaffold adherence, considering the basement membrane preservation of this method. 

Finally, during the maturation or remodeling phase, collagen deposits occur following an organized pattern, which results in a good clinical wound aspect. Successful healing occurs when the new ECM synthesis followed by collagen deposition is superior to the old matrix lysis. Even after one year, the wound will show less organized collagen than healthy skin, and the tensile strength will never return to the native skin, reaching around 80% after three months [61]. Regarding this, considering the period of the fragment’s evaluation in the histopathological analysis, this maturation or remodeling phase was better observed. Macroscopic characteristics of the grafted areas in both decellularized and recellularized grafts showed good aspects considering the remodeling scar and hair follicle presence. Despite the shorter wound closure time in Group III compared to Groups I and II, the healing aesthetic was inferior to those presented by the treated animals. It should be noted that statistical methods were applied for wound closure; however, the quality of healing and tissue recovery, including hair growth, was fundamentally better in Group I and II.

## 5. Conclusions

Our study has verified that both biological decellularized and recellularized grafts are able to induce a healing process in several complex wounds in dogs, with the absence of post-grafting complications and contamination in both groups. In addition, this biomaterial is easily disposable after use and integrates into a less invasive approach for wound healing. The evidence highlighted demonstrates that this biomaterial has the potential to become an innovative therapeutic approach to be applied in the veterinary surgical clinic to restore skin integrity due to severe injuries.

## Figures and Tables

**Figure 1 materials-15-06027-f001:**
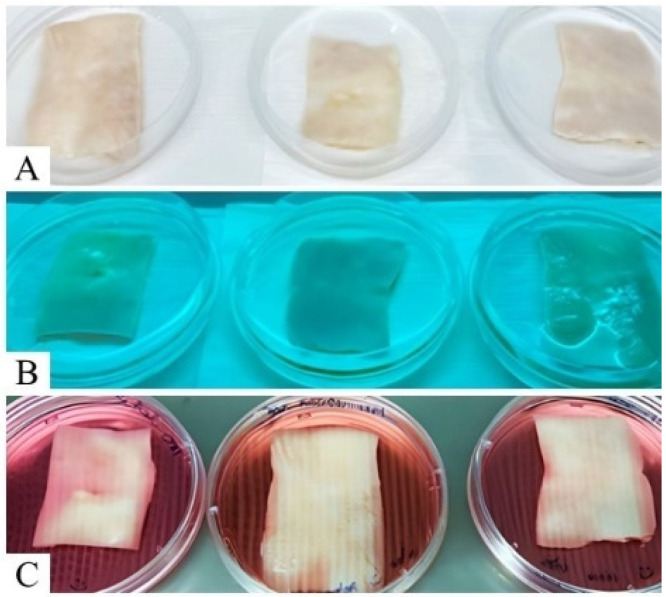
Canine skin fragments before and after decellularization process; (**A**) native skin fragments; (**B**) skin fragments during the process; (**C**) decellularized skin scaffolds.

**Figure 2 materials-15-06027-f002:**
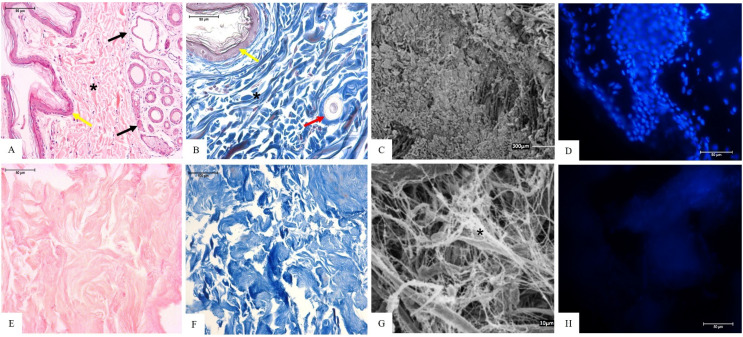
Histology and scanning electron microscopy (SEM) of native (**A**–**D**) and decellularized canine skin fragments (**E**–**H**). (**A**) Native skin tissue, epithelial squamous tissue (yellow arrow), glandular tissue (black arrow), collagen fibers (*), HE staining (Bar: 50 µm); (**B**) native skin tissue, epithelial squamous tissue (yellow arrow), collagen fibers (*), hair follicles (red arrow), MT staining (Bar: 50 µm); (**C**) native tissue, SEM analysis (Bar: 300 µm); (**D**) native tissue, cells in blue, DAPI fluorescence (Bar: 50 µm); (**E**) decellularized tissue, preserved collagen fibers, HE staining (Bar: 50 µm); (**F**) decellularized tissue, collagen fibers in blue, MT staining (Bar: 100 µm); (**G**) decellularized tissue, collagen fibers (*), SEM analysis (Bar: 10 µm); and (**H**) decellularized tissue, DAPI fluorescence (Bar: 50 µm).

**Figure 3 materials-15-06027-f003:**
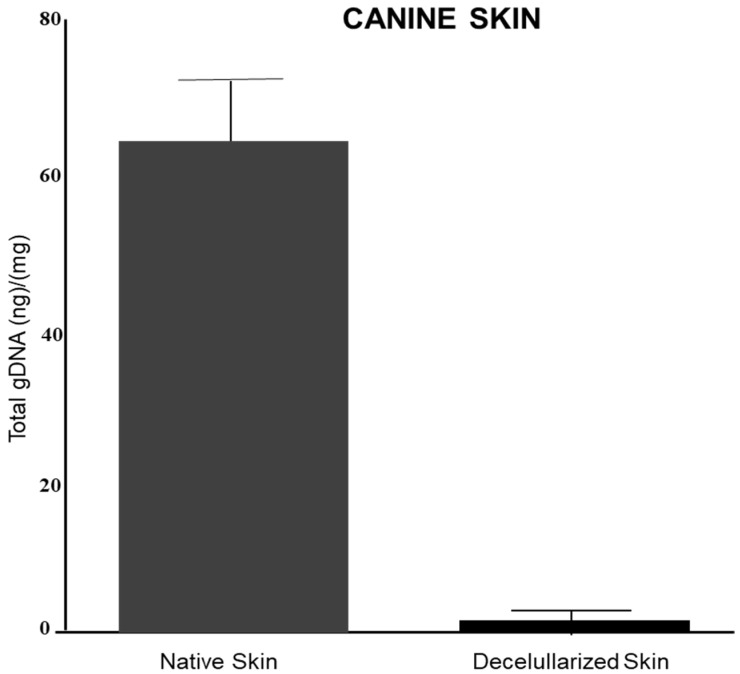
Amount of genomic DNA (gDNA) in nanograms (ng) per milligram (mg) of native and decellularized tissue.

**Figure 4 materials-15-06027-f004:**
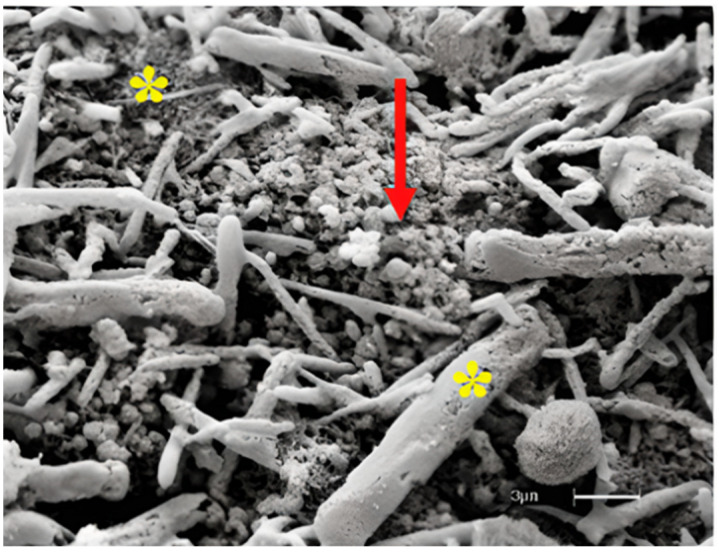
Scanning electron microscopy (SEM) analysis of scaffold recellularized with canine ADMSCs. Note the ECM collagen fiber arrangement (*) and cells attached to the scaffold (red arrow) (Bar: 4 µm).

**Figure 5 materials-15-06027-f005:**
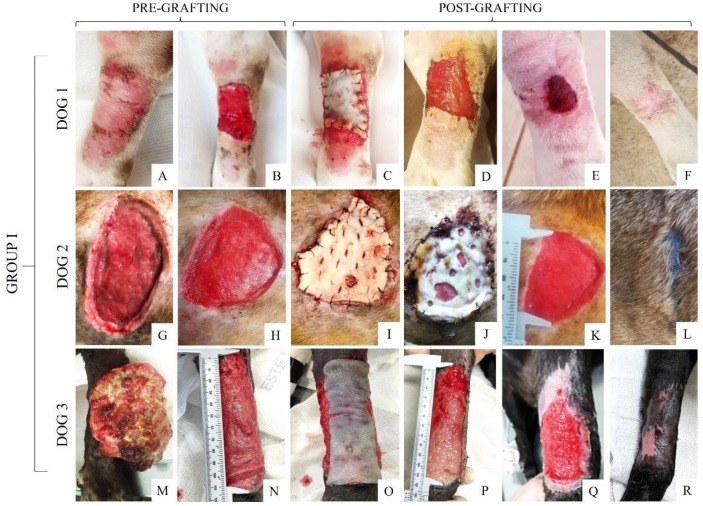
Pre- and post-grafting of the wounds in dogs from Group I. (**A**,**G**,**M**) Initial wound; (**B**,**H**,**N**) healthy granulation tissue 10 days after debridement; (**C**,**I**,**O**) decellularized scaffold grafting technique with complete wound coverage; in (**O**), the image was taken seconds before fixation with standardized suture; (**D**,**J**,**P**) integration of the scaffold into the wound after 48 h; (**E**,**K**,**Q**) clinical follow-up 10 days after grafting; (**F**,**L**,**R**) wound established with 31 days, 42 days, and 45 days.

**Figure 6 materials-15-06027-f006:**
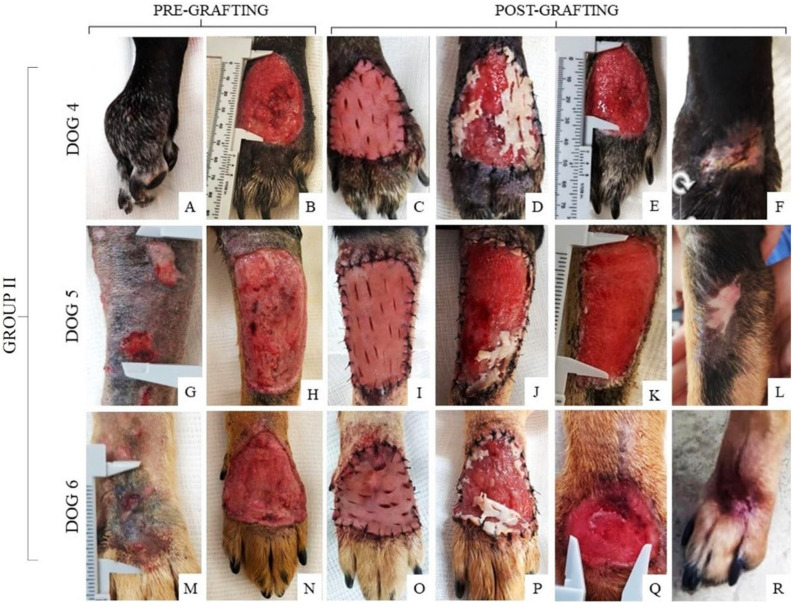
Pre- and post-grafting of the wounds in dogs from Group II. (**A**,**G**,**M**) Initial wound; (**B**,**H**,**N**) healthy granulation tissue 10 days after debridement; (**C**,**I**,**O**) recellularized scaffold grafting technique with complete wound coverage; (**D**,**J**,**P**) integration of the scaffold into the wound after 48 h; (**E**,**K**,**Q**) clinical follow-up 10 days after grafting; (**F**,**L**,**R**) establishment of the surgical wound after 45 days (**F**) and 50 days (**L**,**R**).

**Figure 7 materials-15-06027-f007:**
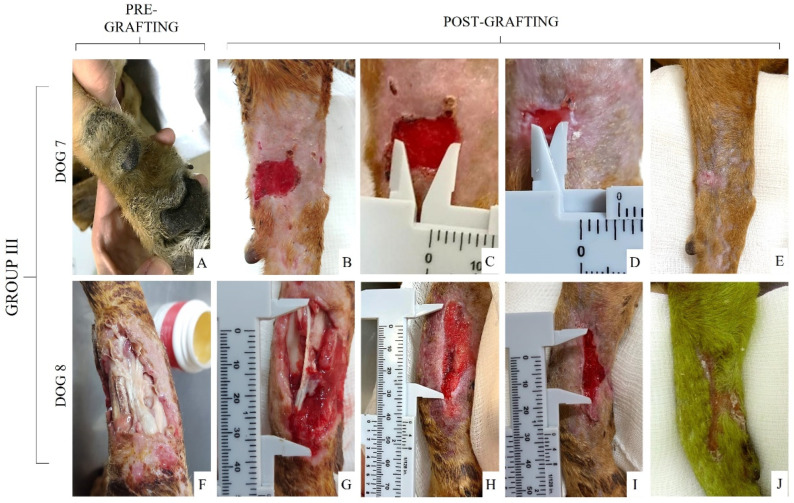
Healing wounds from Group III (control group: wound treatment by second intention). (**A**,**F**) Initial wound; (**B**,**G**) cleaned and healthy granulation tissue after debridement; (**C**,**H**) healing by secondary intention 10 days after debridement; (**D**,**I**) healing evolution 20 days after debridement; (**E**,**J**) establishment of the surgical wound 24 days after debridement and at 35 days.

**Figure 8 materials-15-06027-f008:**
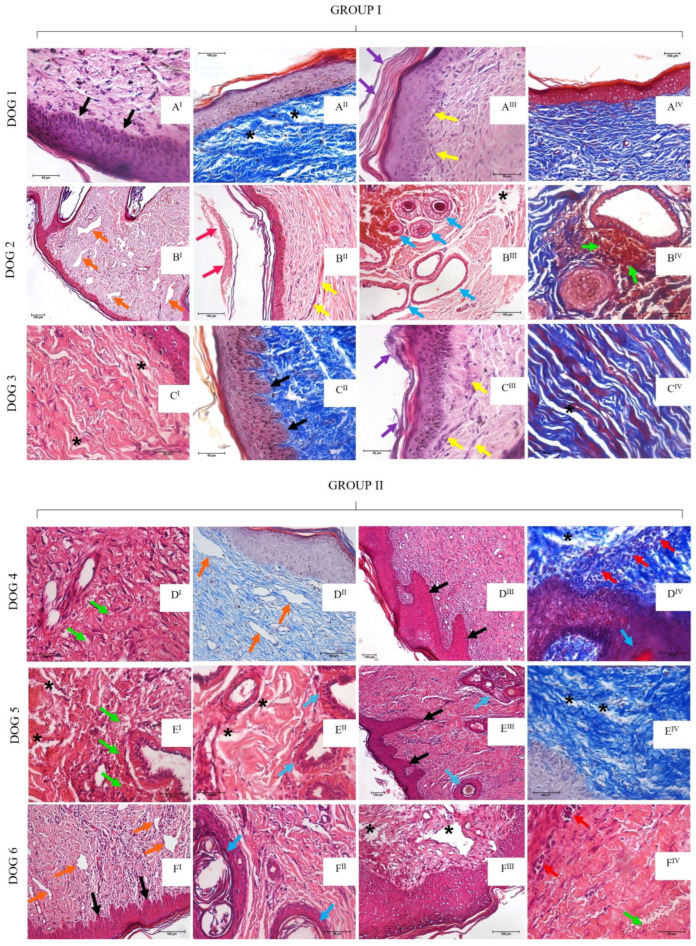
Images from histopathological analysis of Group I and Group II, 30- and 60- days post-grafting. Group I—dog 1: (**A^I^**) (Bar: 50 µm) and (**A^II^**) (Bar: 100 µm)—30 days, HE and TM staining: hyperplasia (black arrows) and edema (*); (**A^III^**,**A^IV^**) (Bars: 50 µm and 100 µm)—60 days, HE and TM staining: hyperkeratosis (purple arrows) and reactive fibroblasts (yellow arrows); dog 2: (**B^I^**, **B^II^**,**B^III^**) (Bars: 100 µm)—60 days, HE staining: angiectatic vessels (orange arrows), fibrinoid material (pink arrows), reactive fibroblasts (yellow arrows), and skin appendages (blue arrows); (**B^IV^**) (Bar: 50 µm)—60 days, TM staining: hemorrhage (green arrows); dog 3: (**C^I^**,**C^II^**) (Bar: 50 µm)—30 days, HE and TM staining: edema (*), melanocytic hyperplasia (black arrows); (**C^III^**,**C^IV^**) (Bar: 50 µm)—60 days, HE and TM staining: reactive fibroblasts (yellow arrows), hyperkeratosis (purple arrows) and edema (*). Group II—dog 4: (**D^I^, D^II^**) (Bars: 50 µm and 100 µm)—30 days, HE and TM staining: hemorrhage (green arrows) and angiectatic vessels (orange arrows); (**D^III^**) (Bar: 100 µm) and (**D^IV^**) (Bar: 50 µm)—60 days, HE and TM staining: hyperplasia and papillary projections (yellow arrows), mononuclear inflammatory infiltrate (red arrows), skin appendages (blue arrow), and edema (*); dog 5: (**E^I^**,**E^II^**) (Bars: 50 µm)—30 days, HE staining: hemorrhage (green arrow), skin appendages (blue arrow), and edema (*); (**E^III^**) (Bar: 100 µm) and (**E^IV^**) (Bar: 50 µm)—60 days, TM staining: hyperplasia and papillary projections (black arrows), skin appendages (blue arrows), and edema (*); dog 6: (**F^I^**) (Bar: 100 µm) and (**F^II^**) (Bar: 50 µm)—30 days, HE staining: angiectatic vessels (orange arrows), hyperplasia and papillary projections (black arrows), and skin appendages (blue arrows); (**F^III^**) (Bar: 100 µm) and (**F^IV^**) (Bar: 50 µm)—60 days, HE staining: hemorrhage (green arrows) and mononuclear inflammatory infiltrate (red arrow). HE: Hematoxylin and eosin staining technique; TM: Masson’s trichrome staining technique.

**Figure 9 materials-15-06027-f009:**
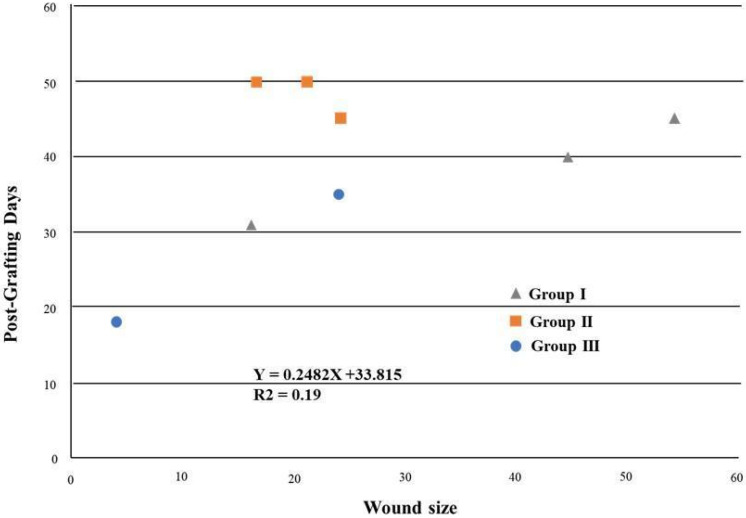
Regression analysis—effect of lesion size on recovery days. Group I and II: decellularized and recellularized scaffold grafting, respectively, *n* = 3. Group III: control, *n* = 2.

**Figure 10 materials-15-06027-f010:**
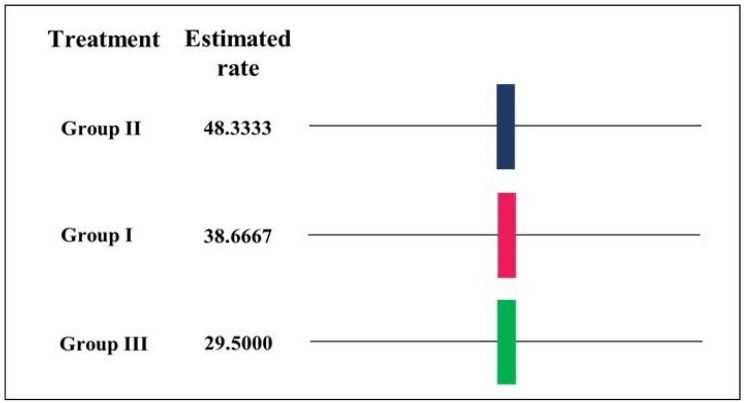
Duncan’s test for means in treatment days (alpha = 0.05). Group I and II: decellularized and recellularized grafting, respectively, *n* = 3. Group III: control, *n* = 2.

**Figure 11 materials-15-06027-f011:**
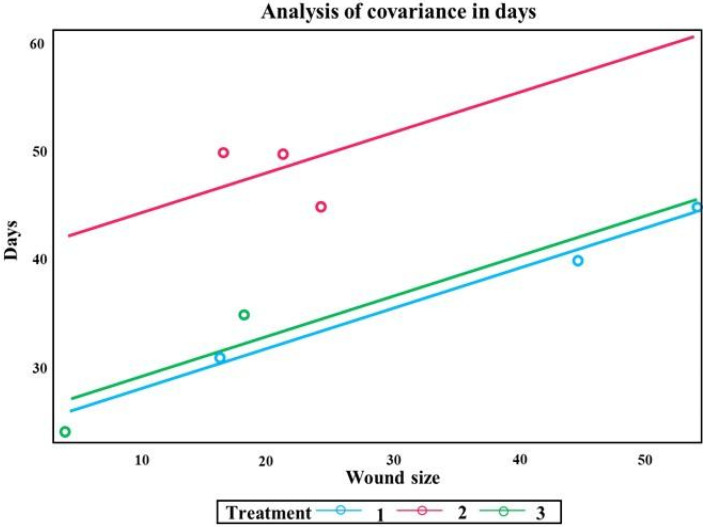
Analysis of variance in days between wound size, the treatment used and healing time. The F statistic for treatment is 17.02 and the *p*-value is 0.0111; for size the *F* statistic is 9.65 with a *p* value of 0.0360.

**Table 1 materials-15-06027-t001:** Description of dogs by group and wound characteristics.

Dogs Description					Wound Characteristics			
	Breed	Sex	Age (Years)	Comorbidities	Type	Cause	Wound Location	Measures
Dog 1 (Group I)	Bull Terrier	M	5	No	Surgical	Actinic dermatitis	Left metatarsal region	4.32 × 3.75 cm
Dog 2 (Group I)	Mixed breed	M	8	No	Bite	Laceration by capybara bite	Right lateral thoracic (costal) region	7.06 × 6.33 cm
Dog 3 (Group I)	Mixed breed	M	12	No	Surgical	Neoformation	Right metacarpal region	12.91 × 4.2 cm
Dog 4 (Group II)	Mixed breed	M	12	No	Surgical	Neoformation	Metatarsophalangeal region, proximal phalangeal region, and right phalangeal region	6.06 × 4.0 cm
Dog 5 (Group II)	Mixed breed	M	7	No	Surgical	Ulcerative dermatitis	Lateral region of left forearm	7.07 × 3.0 cm
Dog 6 (Group II)	Mixed breed	M	7	No	Surgical	Hyperplastic dermatitis associated with dermal fibrosis	Metacarpophalangeal region, proximal phalangeal region, proximal interphalangeal region, and left-middle phalangeal region	4.15 × 4.0 cm
Dog 7 (Group III-Control)	Mixed breed	F	12	No	Surgical	Neoformation	Right lateral metacarpal region	2.3 × 1.75 cm
Dog 8 (Group III-Control)	Mixed breed	F	12	No	Bite	Bite skin laceration	Left plantar metacarpal region	4.9 × 3.7 cm

**Table 2 materials-15-06027-t002:** Mean initial wound size and time required for complete healing among the groups.

Dogs	Initial Wound in cm^2^ (Width × Height)	Time for Complete Healing, Post-Graft Wound (Days)	Average Days, Duncan’s Test (Alpha = 0.05)
Dog 1 (Group I)	4.32 × 3.75 = 16.2	31	38.6667
Dog 2 (Group I)	7.06 × 6.33 = 44.69	40	
Dog 3 (Group I)	12.91 × 4.20 = 54.22	45	
Dog 4 (Group II)	6.06 × 4.00 = 24.24	45	48.3333
Dog 5 (Group II)	7.07 × 3.00 = 21.21	50	
Dog 6 (Group II)	4.15 × 4.00 = 16.6	50	
Dog 7 (Group III)	2.30 × 3.75 = 4.02	24	29.5
Dog 8 (Group III)	4.90 × 3.70 = 18.13	35	

## Data Availability

Written informed consent has been obtained from the patient tutors to publish this paper.

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
