# Peer review of "Biological Graft as an Innovative Biomaterial for Complex Skin Wound Treatment in Dogs: A Preliminary Report"

_materials, 2022, doi:10.3390/ma15176027_

Round 1
Reviewer 1 Report
This study aimed to graft decellularized skin grafts on damaged dog skin to regenerate wounds. Those are some issues in this manuscript that should be addressed before considering the publication in the journal of Materials:
1. According to table 1, the number of samples for each group is so low, and it is hard to analyze statistically? Is there any similar study or article that has been done with this small sample size?
2. In figure 9, R2 is 0.19 (19%), which explains the low possibility of wound healing by this scaffold. How do you explain this challenge? How do you explain it?
3. In terms of the grammar, the manuscript needs to improve such as “extensive wound treatments” instead of “extensive wounds treatments” or “different natural lesions” instead of “different natures lesions”
Author Response
Dear Editor-in-chief
Thank you for giving us an opportunity to submit a revised version of our manuscript entitled “Biological graft as an innovative biomaterial for complex skin wounds treatment in dogs: a preliminary report”
All the reviewer suggestions and corrections addressed are highlighted in red in the main manuscript. The requested changes were made. Moreover, every new modification or rebuttal of the reviewer’s comments is detailed per comment below. We are grateful for the useful comments of the reviewers as a result of which the paper has been considerably improved.
Rebuttal Letter: materials-1832817
Reviewer #1 Comments and suggestions:
Comment 1: According to table 1, the number of samples for each group is so low, and it is hard to analyze statistically? Is there any similar study or article that has been done with this small sample size?
Answer: Dear reviewer, thank you for your consideration. In fact, we understand that the sample number used in our work is low, and a bigger number would be valuable in this case. Unfortunately, considering that our study was applied in animals received at Veterinary Hospital routine, one of our limitations was the access to these cases. In veterinary medicine, clinical trials are limited due to ethical aspects and owners' patients consent. In addition, pandemic also contributed to the lower number of cases attended at the hospital.
A similar study to be cited is Bondioli, E.; Purpura, V.; Orlandi, C.; Carboni, A.; Minghetti, P.; Cenacchi, G.; De Luca, G.; Capirossi, D.; Nigrisoli, E.; Melandri, D. The use of an acellular matrix derived from human dermis for the treatment of full-thickness skin wounds. Cell Tissue Bank. 2019, 20, 183–192, doi:10.1007/S10561-019-09755-W).
In their work, they used a total of 3 human cases, applying acellular skin grafts (osteo-tendineous exposure due to arterial vascular ulcer, bone exposure due to necrotizing fasciitis, osteo-tendineous exposure due to car accident), and this work obtained good clinical results. Cases evolution were analyzed macroscopically and microscopically through conventional histopathological exams; no statistical analysis was applied. The HDM was applied in combination with autologous graft skin on three different clinical cases in which full-thickness skin wounds occurred.
In contrast, even considering the lower number of cases treated in our study, we decided to apply the Ducan test, regression analysis, and covariance analysis, in order to make our study more robust.
Comment 2: In figure 9, R2 is 0.19 (19%), which explains the low possibility of wound healing by this scaffold. How do you explain this challenge? How do you explain it?
Answer: The value of 0.19 (19%) of R2 altered deviation and presented mainly due to the patients of group II, who had the two smallest wound sizes, but that took longer for total healing after grafting. This time was also longer compared to the animals in group I, which despite having larger wounds, had a shorter healing time.
Therefore, we considered two main factors: 1. the presence of cellular material in the recellularized skin scaffold applied for grafting in the animals of group II - based on the histopathological analyses, we suppose that these wounds presented greater mononuclear inflammation, congestion, and edema during a longer period, compared to group I (as mentioned in the discussion), and 2. the nature of the wounds presented by the animals in group II (chronic wounds and neoplasia) - a factor that has now been better presented and discussed in the present revised version of this manuscript, emphasizing and clarifying what is observed in figure 9.
Comment 3: In terms of the grammar, the manuscript needs to improve such as “extensive wound treatments” instead of “extensive wounds treatments” or “different natural lesions” instead of “different natures lesions”
Answer: Dear reviewer, thank you for your consideration. All of them were adopted and the English language of our manuscript was double-checked.
We hope you find the revised version of our manuscript suitable for publication. Thank you in advance for your consideration. On behalf of all authors.

Reviewer 2 Report
The authors have done a commendable job in presenting an interesting article with well defined objectives and conclusions backed by strong data. The writing is clear and straightforward and the material is presented in an organized fashion, and the figures are well-chosen. However, I do have some minor comments/suggestions which might be helpful in improving the article:
1. On page 1, in the last line of the first para, replace "postponing" with "prolonging".
2. Replace "analyzes" with analysis.
3. Table 1 needs to be formatted.
4. On page 8, the last line "The wounds of dogs 4 and 6 had the same healing time" is incorrect as per the images and Table 2, "Dogs 5 and 6 had the same healing time ~ 50 days". Please make the correction.
5. The wounds are not visible on Figure 6A and 7A. Are these the correct figures representing the initial wound images for Dogs 4 and 7?
6. Do a thorough grammar check. Ensure that the spacing is consistent for all sections (eg. section 3.5).
7. Use either dog # or Dog #, instead of using both, to ensure consistency.
Author Response
Dear Editor-in-chief
Thank you for giving us an opportunity to submit a revised version of our manuscript entitled “Biological graft as an innovative biomaterial for complex skin wounds treatment in dogs: a preliminary report”
All the reviewer suggestions and corrections addressed are highlighted in red in the main manuscript. The requested changes were made. Moreover, every new modification or rebuttal of the reviewer’s comments is detailed per comment below. We are grateful for the useful comments of the reviewers as a result of which the paper has been considerably improved.
Rebuttal Letter: materials-1832817
Reviewer #2 Comments and suggestions:
Comment 1: On page 1, in the last line of the first para, replace "postponing" with "prolonging".
Answer: Dear reviewer, the suggestion was accepted. Change made. Thank you.
Comment 2: Replace "analyzes" with analysis.
Answer: Dear reviewer, the suggestion was accepted. Change made. Thank you.
Comment 3: Table 1 needs to be formatted.
Answer: Dear reviewer, the suggestion was accepted. Table 1 was formatted in a horizontal position. Change made. Thank you.
Comment 4: On page 8, the last line "The wounds of dogs 4 and 6 had the same healing time" is incorrect as per the images and Table 2, "Dogs 5 and 6 had the same healing time 50 days". Please make the correction.
Answer: Dear reviewer, the suggestion was accepted. Change made. Thank you.
Comment 5: The wounds are not visible on Figure 6A and 7A. Are these the correct figures representing the initial wound images for Dogs 4 and 7?
Answer: Dear reviewer. The figures are correct. Dog 4 and dog 7 initially presented neoplastic formation - which is represented in these images by the volume increase and local alopecia; although they are not ulcerated lesions (the wound nature of each animal is indicated in Table 1).
Comment 6: Do a thorough grammar check. Ensure that the spacing is consistent for all sections (eg. section 3.5).
Answer: Dear reviewer, thank you for your consideration. All of them were adopted and the English language of our manuscript was double-checked.
Comment 7: Use either dog # or Dog #, instead of using both, to ensure consistency.
Answer: Dear reviewer, thank you. We maintained the word “Dog” in the paragraphs beginning and “dog” in the rest of the text.
We hope you find the revised version of our manuscript suitable for publication. Thank you in advance for your consideration. On behalf of all authors.

Reviewer 3 Report
The work entitled “Biological graft as an innovative biomaterial for complex skin wounds treatment in dogs: a preliminary report” investigates the application of decellularized and recellularized skin scaffolds in different and complex canine dermal wounds. Data reported the perfect integration between scaffolds and the wounds, without rejection and contamination, of decellularized skin grafts. The work is extremely well put together, with a clear introduction of the subject. However, the novelty of this work should be better emphasized. The methodology is clear and very detailed. The presentation of the results is well done; however, many graphics require standard deviation and statistical analysis. Finally, the discussion is pertinent, very complete, and most importantly supported by the literature. I recommend this work’s publication after minor revision – addressing the lacunae referred earlier.
Author Response
Dear Editor-in-chief
Thank you for giving us an opportunity to submit a revised version of our manuscript entitled “Biological graft as an innovative biomaterial for complex skin wounds treatment in dogs: a preliminary report”
All the reviewer suggestions and corrections addressed are highlighted in red in the main manuscript. The requested changes were made. Moreover, every new modification or rebuttal of the reviewer’s comments is detailed per comment below. We are grateful for the useful comments of the reviewers as a result of which the paper has been considerably improved.
Rebuttal Letter: materials-1832817
Reviewer #3 Comments and suggestions:
Comment: The work entitled “Biological graft as an innovative biomaterial for complex skin wounds treatment in dogs: a preliminary report” investigates the application of decellularized and recellularized skin scaffolds in different and complex canine dermal wounds. Data reported the perfect integration between scaffolds and the wounds, without rejection and contamination, of decellularized skin grafts. The work is extremely well put together, with a clear introduction of the subject. However, the novelty of this work should be better emphasized. The methodology is clear and very detailed. The presentation of the results is well done; however, many graphics require standard deviation and statistical analysis. Finally, the discussion is pertinent, very complete, and most importantly supported by the literature. I recommend this work’s publication after minor revision – addressing the lacunae referred earlier.
Answer: Dear reviewer, thank you very much for your consideration. We are excited about all your compliments. In fact, our work is simple and innovative, applying for the first time skin scaffold grafting for wound healing in dogs. We believe this topic is highly useful in Veterinary Medicine. We improved and emphasized the importance of our results in the revised version of our manuscript.
We hope you find the revised version of our manuscript suitable for publication. Thank you in advance for your consideration. On behalf of all authors.

Round 2
Reviewer 1 Report
Thanks for suggested article, I studied it. there was not any statistical study in this study.
A similar study to be cited is Bondioli, E.; Purpura, V.; Orlandi, C.; Carboni, A.; Minghetti, P.; Cenacchi, G.; De Luca, G.; Capirossi, D.; Nigrisoli, E.; Melandri, D. The use of an acellular matrix derived from human dermis for the treatment of full-thickness skin wounds. Cell Tissue Bank. 2019, 20, 183–192, doi:10.1007/S10561-019-09755-W)